# Interprofessional perceptions of emotional, social, and ethical effects of multidrug-resistant organisms: A qualitative study

Stefan Bushuven[1,2,3]*, Markus Dettenkofer[2], Andreas Dietz[1], Stefanie Bushuven[4], Petra Dierenbach[5], Julia Inthorn[6], Matthias Beiner[5], Thorsten Langer[7]

1 Institute for Anesthesiology, Intensive Care, Emergency Medicine and Pain Therapy, Hegau-Bodensee Hospital Singen, Healthcare Association Constance (GLKN), Singen, Germany, 2 Institute for Hospital Hygiene and Infection Prevention, Healthcare Association Constance (GLKN), Radolfzell, Germany, 3 Institute for Didactics and Educational Research in Medicine, Clinic of the University Munich, LMU Munich, Munich, Germany, 4 Institute for Orthopedics, Handsurgery and Traumatology, Hegau-Bodensee-Hospital Singen, Health Care Association District of Constance (GLKN), Singen, Germany, 5 Department of Paediatrics, Neuropaediatrics and Neuro-Rehabilitation Hegau-Jugendwerk Gailingen, Healthcare Association Constance (GLKN), Gailingen, Germany, 6 Center for Applied Ethics in Health Care, Hannover, Germany, 7 Department of Neuropediatrics and Muscle Disorders, Medical Center–University of Freiburg, Faculty of Medicine, University of Freiburg, Freiburg im Breisgau, Germany

* Stefan.bushuven@glkn.de

**Data Availability Statement:** All relevant data are within the manuscript and its Supporting Information files.

## Abstract

### Introduction

Multi-drug-resistant organisms (MDRO) are usually managed by separating the infected patients to protect others from colonization and infection. Isolation precautions are associated with negative experiences by patients and their relatives, while hospital staff experience a heavier workload and their own emotional reactions.

### Methods

In 2018, 35 participants (nurses, physicians, pharmacists) in an antimicrobial-stewardship program participated in facilitated discussion groups working on the emotional impact of MDRO. Deductive codings were done by four coders focusing on the five basic emotions described by Paul Ekmans.

### Results

All five emotions revealed four to 11 codes forming several subthemes: Anger is expressed because of incompetence, workflow-impairment and lack of knowledge. Anxiety is provoked by inadequate knowledge, guilt, isolation, bad prognoses, and media-related effects. Enjoyment is seldom. Sadness is experienced in terms of helplessness and second-victim effects. Disgust is attributed to shame and bad associations, but on the other hand MDROs seem to be part of everyday life. Deductive coding yielded additional codes for bioethics and the Calgary Family Assessment Method.

**Funding:** The Messmer Foundation only covers the publication costs only. The whole study itself was conducted without external financial support. The funders had no role in study design, data collection and analysis, decision to publish, or preparation of the manuscript.

**Competing interests:** The authors have declared that no competing interests exist.

**Abbreviations:** HCP, Health Care Provider; MDRO, multidrug-resistant organisms.

## Conclusion

MDRO are perceived to have severe impact on emotions and may affect bioethical and family psychological issues. Thus, further work should concentrate on these findings to generate a holistic view of MDRO on human life and social systems.

## Introduction

In this manuscript we report on interprofessional post-graduate health care workers' perceptions of the emotional, psychological and ethical effects provoked by multidrug-resistant bacteria (MDRO) and isolation precautions.

Multi-drug-resistant organisms have a significant impact on patient safety, in-hospital mortality and the economic burden [1–3]. Especially carbapenemase-producing enterobacteria (CPE), methicillin resistant staphylococcus aureus (MRSA) and vancomycin resistant enterococci (VRE) are known for outbreaks or severe courses of infectious diseases. Control strategies include hand and surface hygiene [4], barrier precautions (gloves, gowns, face-masks) including isolation [5]. Unfortunately, as such barrier precautions have proven to be ineffective on their own, they are questioned [6] in favour of hand hygiene and antimicrobial stewardship programs [7].

These restrictive isolation protocols or even intimidating equipment (masks, gloves, gowns) may provoke emotional reactions in health care providers and patients [8]. Emotional phenomena like anxiety, anger, sadness, disgust and enjoyment [9] are core behavioral 'programs' on stimuli, context and volition innate to every human and even animals [10, 11]. They are distinguished from 'higher' feelings (e.g. pride, honor, hate) [12]. Emotions serve as rapid social and psychological responses in human interaction [13] and play a significant role in the patient-physician and patient-nurse relationship [14–18].

Although there is growing evidence about the psychological side effects of MDRO on patients [16, 19–23], little is published about the holistic impact on patients and their families or on health care providers themselves. One model to describe comprehensive issues concerning families is the Calgary Assessment and Intervention Model (CFAM) [24]. This model comprises six main factors with some examples added for infectious disease:

- structural context (ethnicity, race, social class, religion, and spiritual aspects–e.g., infections as a divine punishment [25]),

- external structures (friends, relatives, networks–e.g., friends not visiting the MDRO patient because they fear becoming infected [26]),

- -internal structures (inner family composition, gender, rank–e.g., responsibility for family members [23]),

- -functional-instrumental (daily activities–e.g., self-isolation [27]),

- -functional-expressive (communication, problem-solving, beliefs, powers–e.g., family guided decolonization [28]) and

- developmental (future, plans–e.g., amending future plans because of MDRO) contexts.

As shown above, recent management options comprise limiting visits and mobility impairing autonomy and free will. Beyond this, malevolent stigmatism [29] as an infectious patient

may provoke fear in visitors and family members. Barrier precautions and isolation without a clear and proven benefit for the affected person are not benevolent and raise issues of justification for those concerned. In contrast, persons with no MDROs may fear resistant bacteria and demand restrictions for those colonized or infected. This leads to an ethical dilemma requiring justification for the decisions made. Complementary to the CFAM, these four aspects (autonomy, benevolence, malevolence and justice) represent the principles of bioethics described by Beachamps and Childress [30]. Put together, MDRO can have both an impact on emotions and feelings as well as affect family factors and ethical decisions on questioned [6] and restrictive MDRO management.

## Previous work

In 2018 our working group evaluated subjective views of multiprofessional care givers about how they experience patients' and other health care workers' reactions on isolation precautions.

In this study [8] we used facilitated discussion groups consisting of post-graduate health care workers from 11 different professions in an educational setting: Participants in this study were given two cards with a medical profession (other than their own) written down on it. They were expected to write down their impressions and experiences concerning how the professionals on the card react to MDRO. These results were pinned on a cardboard and discussed in the group. The cards were inductively coded by our working group consisting of three coders. After showing a relevant association with emotional aspects in the first coding round, we conducted a second deductive round using the set of the main emotions, described by Ekman et al. [9]: anger, disgust, anxiety, enjoyment and sadness. Considering this framework, the following themes were evaluated, showing health care providers experiences with MDRO management:

- Patients are admitted to hospital for high quality treatment of a severe and threatening illness or condition like coronary heart disease, cancer, or neurological disease.

- If confronted with MDRO, patients and relatives need further medical information.

- Too little knowledge and time or inability to communicate knowledge to patients lead to the lack of, and/or inconsistent and contradictory information.

- A lack of knowledge on the part of health care providers may lead to over- and underestimating risks.

- For patients and their relatives, contradictions in MDRO management provoke uncertainty, anxiety and anger.

- Combined with a severe underlying medical condition this aggravates fear and stress due to the need for time-consuming consultations, taxing valuable critical resources in the daily routine.

## Scientific issues

Keeping our previous study in mind the follow-up described in this manuscript focuses on emotions applying a similar study design, but addressing specifically the emotional effects on patients, caregivers, and hospitals in conjunction with MDROs.

Our main hypothesis a priori was that inductive findings of the first study could be proven again in health care workers and may lead to a deeper insight on emotional impact than detectable in the first study. This article provides readers with our inductive and deductive analysis

of a set of facilitated discussion groups made up of post-graduate health care providers of different professions und from several hospital-departments of anesthesiology, emergency medicine, critical care, pharmacy, and medical education. We report on the educational, psychological, and ethical perceptions of these three groups about MDRO-effects on patients, families and staff.

## Materials and methods

Material and methods are presented according to the COREQ-checklist.

According to the ethical committee Stuttgart there was no need for further ethical approval. We included the decision letter of the ethical board as supporting information S1 File.

### Study design

In this qualitative study, we engaged six facilitated discussion groups each containing 5 to 10 post-graduate medical professionals in service recruited from six hospitals (three primary, two secondary, one rehabilitation unit), and 11 medical educators from the Master of Medical Education course forming the seventh group for external validation. Hospital discussion groups were embedded within a six-hour interprofessional training session for postgraduate health care providers focusing on antimicrobial stewardship (AMS) in the German language in 2018. All participating trainees were enrolled in this study. Except for underage, there were no other inclusion or exclusion criteria for participants representing in-service medical professionals from these hospitals to participate in a learning program. None of the participants prohibited the use of their qualitative data derived from the lessons.

According to regulations from the ethics board and workers' council further quantitative data aside professional groups was not obtained to guarantee anonymity. We aimed to enhance our findings' generalizability by taking interprofessional and qualitative approach. We thus recruited participants from six hospitals and an external group of professionals to cross-validate our data. To guarantee anonymity, informed consent was obtained orally at the beginning and end of each course. Participants were informed about the use of anonymous data for scientific research during the registration process and the possibility to refuse or withdraw participation without incurring any disadvantage.

The learning format here, differing from our original study's [8] was developed applying active learning methods [31] and curriculum development tools [32] and was open to physicians, critical care nurses, and pharmacists. It was divided into three lessons, each consisting of a passive part (short report lasting 20 minutes), an active part of different format (about 90 minutes) and a reflective part (card survey, about 10 minutes). These study data were collected in the first active part. The complete course is illustrated and described in detail in Fig 1. In the active part, 35 participants were exposed to five statements and were expected to write down their experiences, impressions, and thoughts on cards (Fig 2). The five statements were "MDRO leads to . . ." ". . .Anger", ". . .Fear", ". . .Enjoyment", ". . .Disgust" and ". . .Sadness". Participants had up to 15 minutes to write down their impressions and experiences on cards, followed by discussion. The moderator (facilitating the process, guiding the participants and when asked, answering other questions) took an active part as in a socio-constructivist (e.g., asking about details, clarifying answers) approach. The 11 participants in an external medical-education course fulfilled a validation purpose for us to seek for further tags and codes to ensure data saturation. For these, we applied the same technique of the facilitated discussion alone without additional training elements.

Approval from the ethics board of the Stuttgart Physicians' Association was obtained prior to this investigation.

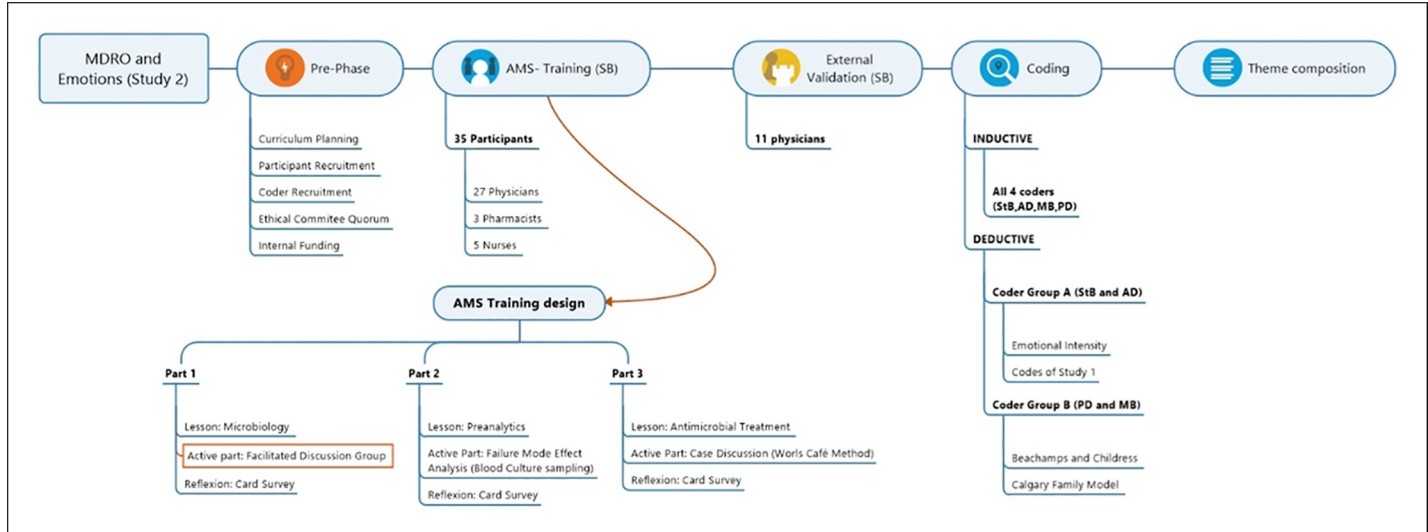

**Fig 1.**

### Research team and reflexivity

The main investigator of this study and facilitator is a 41-year-old male senior consultant anesthesiologist, critical care, and emergency physician, hospital hygiene & AMS expert and medical educator (MSc in medical education) with international certification as a medical risk

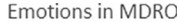

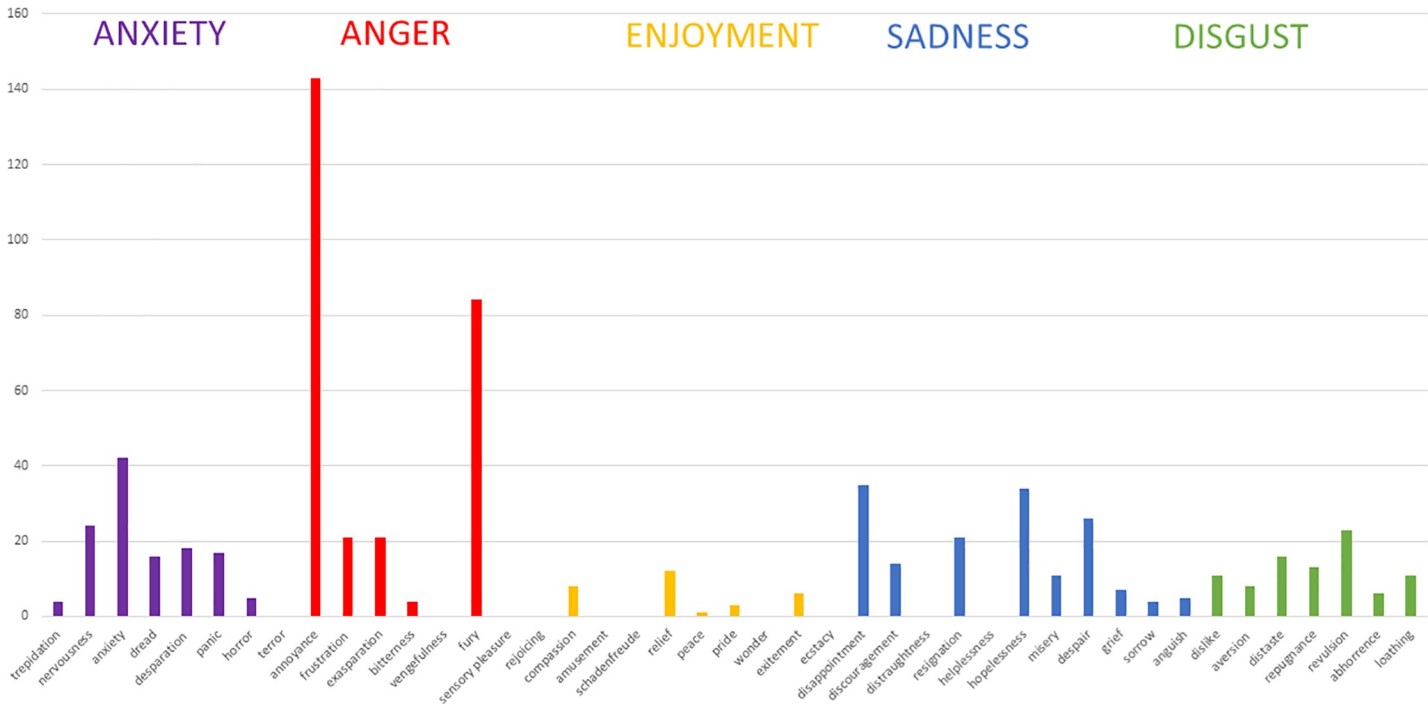

**Fig 2.**

manager. His main responsibilities at the time of this project were infection prevention and multiprofessional medical education (under- and postgraduate physicians, nurses, and anesthesiology assistants). The study supervisors were a male consultant (PhD) in neuropediatrics experienced in interprofessional medical education and qualitative research, and a male hospital hygiene expert (PhD, associate professor, head of department of the hospital-hygiene department) with practical experience in environmental and healthcare infection prevention. Coders were a male anesthesiologist with a master's degree in design, a female social education worker specialized in systemic family therapy, a male resident in pediatrics and a female trauma surgeon all with experience in interprofessional education Two coders had worked in our preceding study. Coders were instructed in the coding procedure using individual- and team- reflections and supervision by the investigators.

About 40% of the study participants were known to the main investigator before the course. Ten percent had been in regular contact with him.

**Analysis.** The main investigator as a single researcher [33] analyzed the data using a iterative inductive approach with 1) assignment of simplified "tags" to recorded items, 2) development of specific "codes" from these tags (recontextualized results and meanings) and 3) deriving themes (recontextualization and comprehension of results) [34]. In contrast to the first study, he did not participate in code allocation.

All coders participated in coding the inductive findings. For these, a code was accepted if three coders allocated a code to a tag. After acknowledgment of many "ethical" and "psychological" codes, two coders concentrated on the second round of deductive coding applying the four principles of biomedical ethics according to Beauchamps and Childress [30] on the one and the Calgary Intervention Model [24] on the other hand. The other two (both involved in the preceding study) coded additionally in line with our prior study's findings [8]. In both deductive approaches, two coders had to agree to assign a code. Additionally, we took a semi-quantitative approach to illustrate our results and the frequency of code allocation (see Table 1).

Codings were analyzed using MS Microsoft Excel (Microsoft Corporation, Redmond, USA). Illustrations were done using MindManager 2019 (Corel Corporation, Ottawa, Canada).

## Results

### Participants' characteristics

All 35 participants (27 ED and ICU physicians, 3 pharmacists, 5 ED and ICU nurses) of the hospital groups were recruited from six hospitals in Lake of Constance Region. Recruitment was done via the academy of health care professionals in Singen am Hohentwiel. Six courses were held from April 2018 to November 2018 at the Hegau-Bodensee-Klinikum Radolfzell. The external validation group (11 physicians all from different hospitals in Germany) was assembled in June 2018 and was recruited from a medical education event at the University Hospital Bonn.

### Coding process

A total of 773 items were decontextualized (214 for anger (27.8%), 208 for fear (26.9%), 136 for sadness (17.6%) and 127 for disgust (16.4%), 88 enjoyment (11.4%)), tags (comprehension and of items) allocated, and codes (meanings) and themes and subthemes formed.

We assigned 10 inductive codes for anxiety 10, 11 for anger, 7 for joy, 5 for sadness and 8 for disgust (see Table 1) forming each category's sub-themes.

We identified three main-themes and 14 sub-themes.

**Table 1. Codes and main themes according to inductive codings.**

| | Inductive Code | Sub-Themes |
|---|---|---|
| ANGER (214) | 1 Commercial aspects & greed (8) | 1) FAILURES OR INCOMPETENCE |
| | 2 Economic loss (6) | 2) WORKFLOW IMPAIRMENT |
| | 3 Patients' irrational expectations (36) | 3) KNOWLEDGE |
| | 4 Unfairness (7) | |
| | 5 Deterioration of prognosis (37) | |
| | 6 Disruption of workflow (36) | |
| | 7 Failing communication (61) | |
| | 8 Lack of Empathy (41) | |
| | 9 Helplessness (10) | |
| | 10 Lacking resources (22) | |
| | 11 Lack of education and knowledge (31) | |
| ANXIETY (208) | 1 Difficulty to provide information (9) | a) KNOWLEDGE |
| | 2 Fear concerning to be responsible (23) | b) RESPONSIBILITY |
| | 3 Lack of information (101) | c) ISOLATION DESPITE NEED FOR SOCIAL HELP |
| | 4 Helplessness towards MDROs (44) | |
| | 5 MDROs are part of everyday life (14) | d) BAD PROGNOSIS |
| | 6 Failing knowledge produces fear (89) | e) MEDIA RELATED EFFECTS |
| | 7 Separation and social isolation (27) | |
| | 8 Deterioration of prognoses (26) | |
| | 9 MDROs & Media: epidemic threat (9) | |
| ENJOYMENT (88) | 1 Scientific fascination (14) | a) A MEAN TO AN END |
| | 2 MDROs can be used as a tool (16) | b) OVERCOMING |
| | 3 Relief to know one was not responsible (3) | |
| | 4 Happiness to have avoided negative situations (3) | |
| | 5 Economic benefit (8) | |
| | 6 Possibility of target control therapy (12) | |
| | 7 Victory over MDROs (20) | |
| SADNESS (136) | 1 MDROs produces helplessness (32) | a) HELPLESSNESS |
| | 2 Death and Illness (25) | b) SECOND VICTIM |
| | 3 Fear of financial difficulties (3) | |
| | 4 Second Victims (2) | |
| DISGUST (127) | 1 MDROs is an epidemic (35) | a) SHAME of being infected |
| | 2 MDROs are equated to uncleanliness (41) | b) PART OF EVERYDAY LIFE |
| | 3 People perceive MDROs differently (25) | c) ASSOCIATIONS caused by MDRO |
| | 4 Gloves and gowns minimize feeling disgusted (8) | |
| | 5 Self-disgust (15) | |
| | 6 Associations (3) | |
| | 7 Part of everday-life (2) | |
| | 8 Sensory experience (10) | |

The numbers in the first column indicate the total number of allocated items. Each number in the second column indicates how often 3 or 4 coders codified that code.

## Main-theme 1: Verifying the emotional impact of MDROs

**Anger.** Participants reported expecting anger in themselves and in patients in different situations: The first subthemes were that MDROs are caused by a personal ***experience of*** failures, an accident or even ***incompetence*** that needs to be punished (*"They did not manage it. Now everything is lost"*, *"They are responsible for that!"*) putting patients at further risk (*"The germ*

must be killed!”), leading to inequality (“*Isolation is unfair*”, “*The patient must be the last one in the operation theatre*”) and to the experience of care givers’ helplessness (*“Our precautions failed to work”*, *“Why are they not successful?”*). The second showed that **MDROs impair the hospital workflow** by intensifying staff shortages, provoking inter- and intraprofessional conflicts, and compromising the quality of medical care (“*The nurses keep annoying me about the MDROs”*, *“It’s annoying”*, *“More work, less time”*, *“It’s bothersome. Your ability to work is impaired”*, *“I don’t understand why the isolation rules do impair my work!” “I can’t work like this. I am frustrated!”*, *“A pregnant colleague won’t go in the room. She leaves everything for me to do”*, *“Diagnostics are delayed”*, *“More effort. Bad care. Slower patient recovery”*, *“You have to re-schedule everything. The entire ward organization is messed up”*). Anger was attributed to external factors, but not to one’s own work with MDROs in all items.

**Anxiety.**    Concerning anxiety, we differentiated between fear and anxiety. Fear (specific, short-term, seeking immediate reactions) was coded 15 times, while anxiety (unspecific, long-term, future-oriented, seeking safety) was coded 79 times. Participants reported about being dependent on information to relief **anxiety**. Care givers perceive anxiety when having to inform patients about MDROs “*Information to patients provoke anxiety and fear if you tell them, what could happen*” and that “*Anxiety is information-dependent*” especially when patients cannot comprehend, what their MDRO is meaning to them (“*What is happening to me?*” “*What will happen in future?*” “*Will I lose my leg?*”) or to their families (“*What happens now to my family when I come home?*”). **Inconsistend information**, failed information transfer, inconsistent and contradictory information cause patients to feel anxious (“*Patients think they will directly catch the bugs in hospital*”, “*The greater the information deficits, the bigger the panic*”), visitors (“*So they freak out if I touch anything!*”) and staff (“*When will the operation room cleaners come? Will they come in this case?*”, “*Nurses flip out in the recovery room*”, “*One have no control if that, what we explained about the bugs was really understood*”).

This is linked to phrases health care workers were confronted with showing patients’ **fear of being socially isolated** or even abandoned in a time, patients are in need of their families and relatives: “*Stigmatism*”, “*My loved one is not being cared for*”, “*It’s like leprosy*”, “*Fear of touching the family member*”, “*Patient fear to be isolated*” “*You have got the bug. We must isolate you!–And what happens to me?*” “*The nurses take all isolation precautions, but the patient is the last to hear about it, sometimes they get out of their room and notice the gowns and the isolation signs. That is a harrowing experience!*”

Nevertheless, MDRO are known to medical staff and patients to reduce the **prognosis** of the underlying disease or may lead to additional disease leading to anxiety: “*Bad prognosis*”, “*Permanent damage*”, “*Longer hospital-stay, sometimes death*” and “*Reduced prognosis especially in the multimorbid patients*”, were mentioned multiple times. For medical staff, experiencing MDRO as a routine phenomenon, **responsibility** and anxiety to be blamed for MDRO-linked conditions like “*Bad conditions*”, “*Death*” “*Impairment*”, “*Loss of limbs*” were mentioned several times.

After all, MDRO related fear is experienced to be **facilitated by the media**, leading to more dis-information and to fear and anxiety on the side of patients and staff, fearing to be blamed: “*The press creates panic about the mortal danger from MDROs*”, “*TV and media cause fear*”, “*The media cause it. One mistake can make everything unhygienic.*”

**Enjoyment.**    Paradoxically, mentioning this emotion together with MDROs showed to amuse most participants, as they wrote a “*No*” on their cards. Some participants mentioned that they experience it as **a means to an end** for scientists, microbiologists, pharmacists and media: “*Scientist are curious about MDRO*”, “*These bugs fascinate biologists*”, “*Reporters earn their money from them and bad news*”, “*The drug industry is happy about it, so they can invent something having the reason to raise profits*”. Further the **conquering of MDRO** seems to make

everybody happy: "*When it's gone*", "*When you can deliver the good news that isolation has ended*", "*When your team overcomes the hurdles caused by MDRO*" and "*If you manage to treat your patient successfully and beat the bugs*".

**Sadness.** Sadness was perceived to be present for both patients, relatives, and staff. **Helplessness**, frustration, and resignation were coded several times showing the subjective overwhelming force of MDROs concerning medical and economic aspects: "*With open eyes into perdition*", "*There is nothing left, we can do for these patients*", "*There is nothing we can do for our family member*", "*There is not hope.*", "*You know, you can die from these bacteria. And you know, there is not much left we can do.*" were typical phrases. Furthermore, **Second Victim Effects** [35] and disappointments with compassion and a feel to be guilty or to have failed were mentioned: "*We did not inform the patient correctly*", "*Malpractice*?", "*The success of the operation vanishes in the face a MDRO wound infection*".

**Disgust.** Disgust yielded three main sub-themes: First, participants mentioned, that patients feel **ashamed** of being tested "positive" and being infected, feel self-loathing and therefore to be "*dirty*", "*filthy*" and stigmatized: "*Ugh, he's caught the bug!*", "*If a patient in gloves and gowns enters the cafeteria, you feel disgust.*" "*You see the gazes of the people, if they see an isolated patient.*", "*You do not want to touch anything the isolated patient had in his room before*", "*Patients feel disgust concerning themselves.*" "*Everybody evades a patient in gowns, if you enter the elevator*". On the other hand, and for health care providers most participants stated, that MDRO are **part of everyday life** but may provoke age-depending disgust in case of "*purulent wounds*", "*extreme smells*", "*seeing diarrheic stool*", "*bacterial growth in wounds like pseudomonas*", "*You feel smelly by yourself, if you take of the gowns and gloves after caring for a MDRO patient*" especially for "*the younger care givers*", "*students*" and "*young doctors*". Third, disgust may be provoked by **associations** linked to MDRO: "*It is disgusting when you see a patient with new VRE, and you know, that his neighbor days ago had it in his rectum*", "*You associate it with filth and plague.*", "*It's disgusting, when you think about hygiene mistakes and errors in general*",

**Intensity of emotions.** most emotions were distributed over the full spectrum of intensities (see Fig 2). The exception to that is that "fear" revealed mainly low-intensity codes, whereas "fury" revealed low intensity (i.e., feeling bothered or annoyed) as well as high intensity (rage, seeking revenge).

## Main theme 2: MDROs are linked to ethical considerations

In the first deductive part, all items were coded using the four principles of medical ethics (see Fig 3). These showed that **benevolence** and **non-maleficence** were coded to a large proportion of items linked to less care and giving up patients: "*last on the operation schedule*", "*no information to patients*", "*information provokes fear*", "*wrong consequences*", "*the docs are no longer seeing the patient*", "*these patients are uninformed*", "*stigmatization*", "*patients think they are going to die*", "*nobody is visiting them*" and "*neglect of patients*" were some of the phrases noted. In contrast, **autonomy** was seldom coded ("*longer stay in hospital*", "*I feel compassionate towards the patients, because they are not allowed to leave their room*", "*The patients are locked into their room*") and mostly related to patients' restricted freedom, due to being isolated. Justice codings was seldom and mainly focused on the subjectively perceived unfairness of the workload's distribution ("*more work for us–and there is no help*") and economic aspects ("*economic loss*", "*MDRO is expensive*", "*beds have to be blocked*").

## Main theme 3: MDROs impaired family functioning

Six deductive codes (3 structural, 2 functional and 1 developmental) according to the Calgary Family Intervention Modell by Wright and Leahey were assigned to all items (see Fig 4):

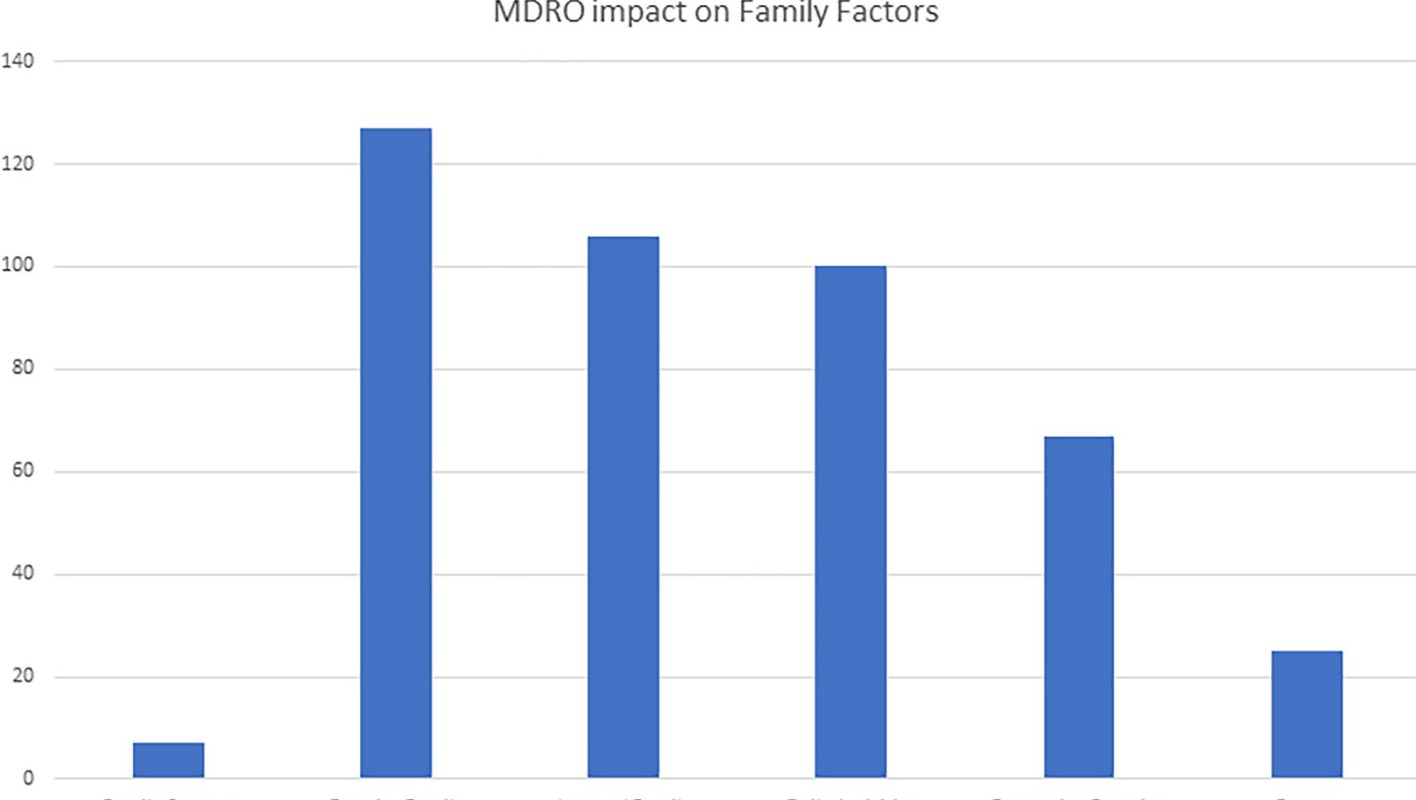

**Fig 3.**

**Structural—Contexts** (ethnicity, race, social class, religion, spiritual) were coded only seven-times and were associated to knowledge and backgrounds of families. In contrast, 127 items were coded for the **Structural-External** (larger family composition, relatives, friends, and other networks). Most codes for emotions were linked to being socially isolated from friends and to shame and the fear of transmitting MDRO. **Structural-Internal** codes (family composition, gender, hierarchical rank, subsystems) yielded similar findings, like problems with knowledge ("*Relatives don't understand MDRO management*", "*No knowledge leads to panic*") leading to "*irrationality*", fear and despair ("*O my God–what will happen to my children*?", "*Will I ever get to go home*?"). Comparable, functional codes revealed severe impairment of **functional-instrumental** (daily activities) and **functional-expressive** (family communication, problem-solving, beliefs, power) aspects witnessed in patients and families with MDRO: "*They do not know, how to manage MDRO*", "*helplessness*", "*stigmatization*", "*social isolation*", "*loneliness*" were mentioned in conjunction with anxiety and sadness. "*being blamed*" of different types and intensity ("*malpractice*", "*guilt*", "*being prosecuted*", "*getting even*") were coded for anger several times. **Development** items (tasks, plans) were coded only 25 times. Those codes were mainly associated with to fear and anxiety concerning MDRO management and uncertainty.

### How themes from the last study were deductively coded

Finally, all inductively derived codes in our first study was coded deductively in our findings to confirm our primary study's results. All codes, namely lack of knowledge on the part of

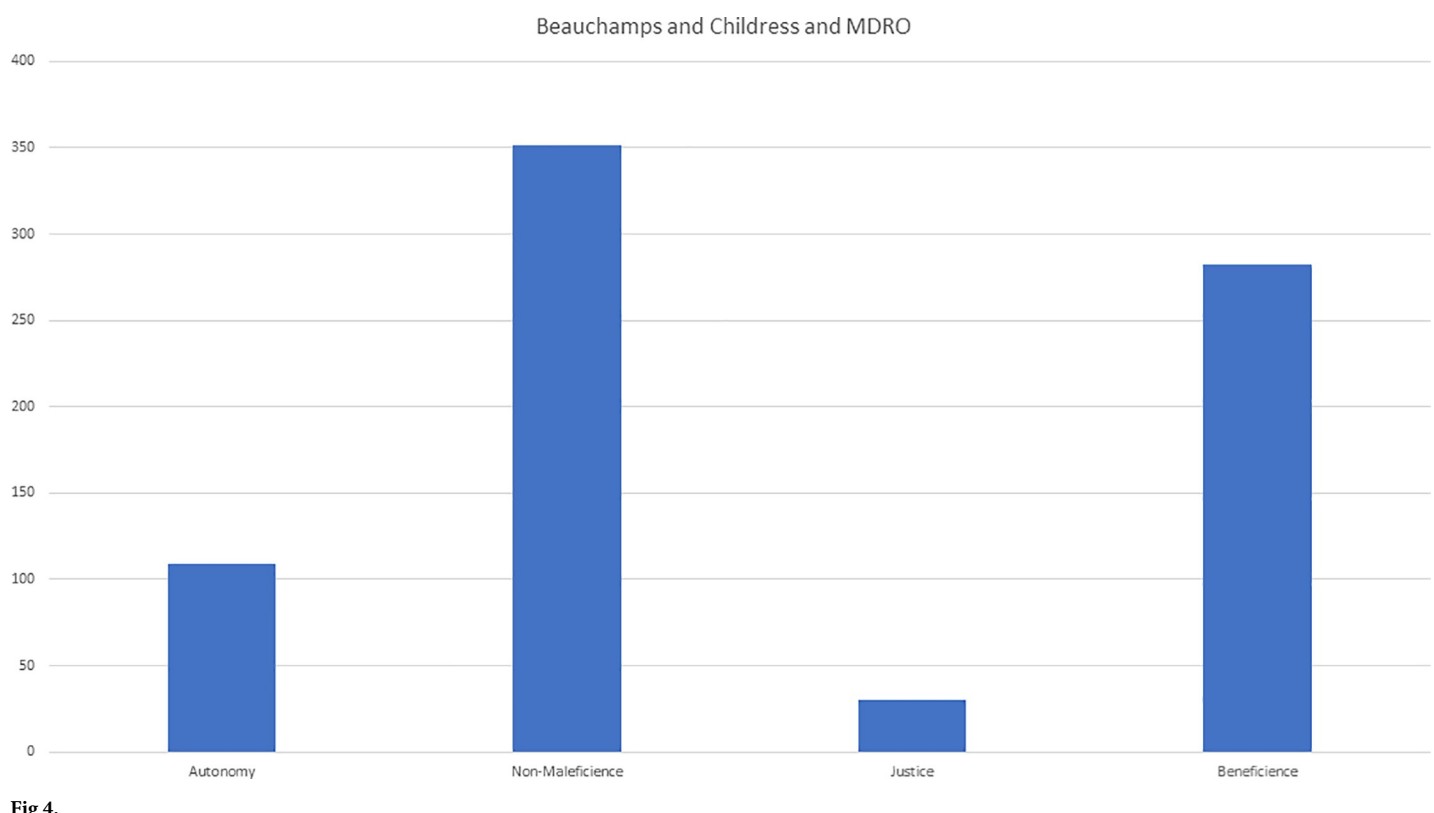

**Fig 4.**

specialists and lay persons, insufficient information transfer, lack of resources, impairment of prognosis, over-rating the MDRO leading to excessive isolation and panic, under-rating the MDRO leading to bacterial spread, poor communication techniques and inconsistent information within medical staff and globally, were coded several times.

## Discussion

Relying on the experiences of health care providers this study demonstrates interactions between MDRO management, medical and general information, and emotional factors. We also demonstrate, an association with medical ethics and family psychological factors, which were acknowledging the subjectivity of personal observations. We conducted this investigation specifically to compare these findings to the preceding study's [8]. We hope to inspire discussion and novel hypotheses (see Fig 5):

### 1. MDRO cause an imbalance of information aggravated by and resulting in emotional reactions

According to the perceptions of health care workers our results show, that a MDRO diagnosis can overstretch the medical knowledge of health care providers, who complain about staff shortages and time pressure [36]. Caregivers' perceptions of not having been adequately trained in MDRO management [8, 37] are also associated with their emotions (anger about a heavier workload due to barrier precautions, anxiety about contracting the MDRO or endangering their family, feeling sadness as compassion for the MDRO patient, disgust among younger staff), over- or underrating the MDRO risk, and the inability (due to lacking knowledge-transfer know-how [38, 39] or time pressure) to provide patients with valuable information.

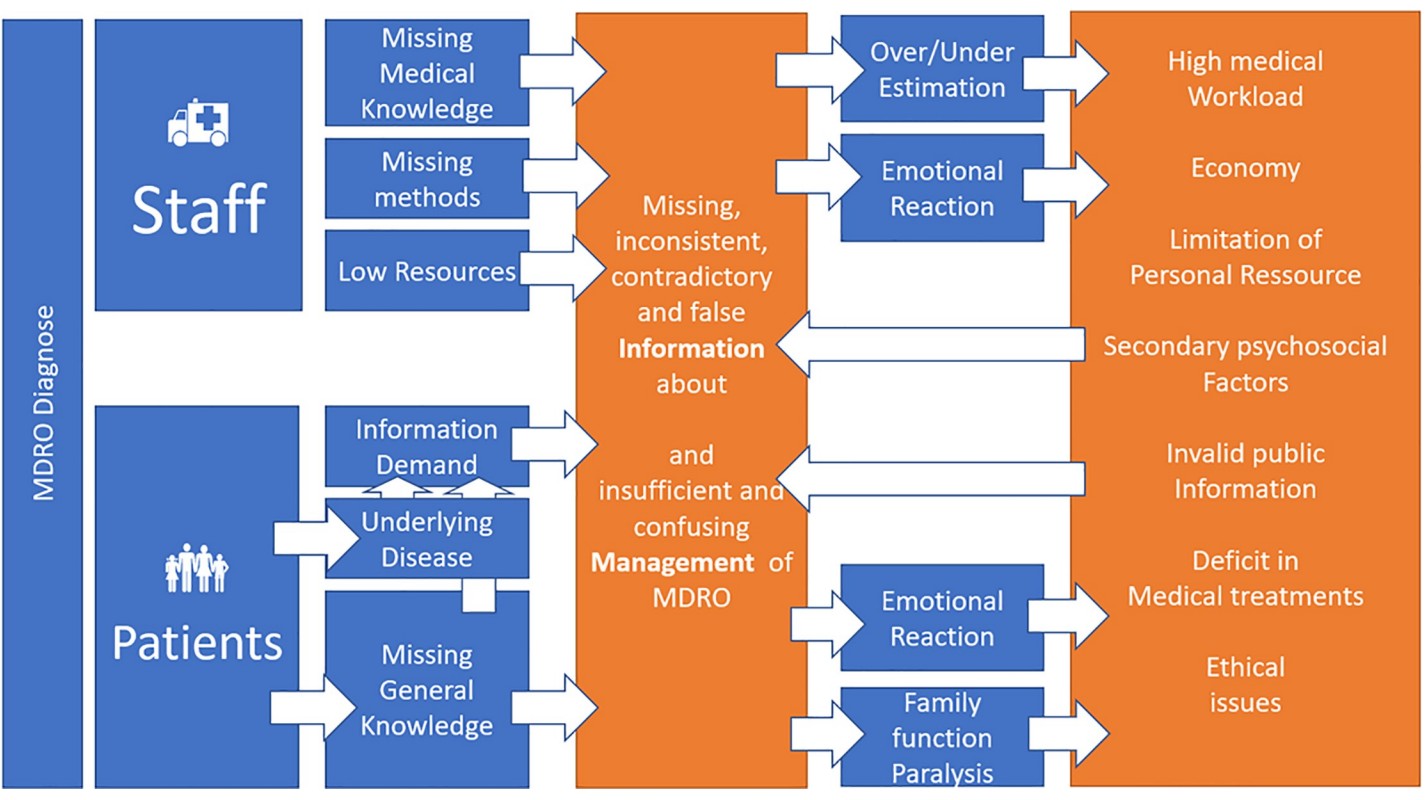

**Fig 5.**

The discussion groups' most frequently expressed emotions were especially anger and anxiety, and even the "enjoyment" items yielded items suspicious of triggering anger and revulsion toward external groups (i.e., scientists, reporters, hygiene experts), whom the caregivers participants thought might profit from MDROs.

Participants reported that patients and their relatives need information especially if there is a threatening underlying medical condition [14]. Medical staff having limited information and resources and with their own emotional reactions to MDROs, fail to transfer information, so that patients and their families feel helpless and unable to control the situation. Caregivers feel anger, sadness and despair as a consequence [40] and having to handle increased workload to compensate for these emotions to secure or reestablish the relationship between care givers and patients is experienced [18].

## 2. MDRO impairs family functioning

The deductive codes we derived applying the Calgary Family Intervention Model [24] show that health care providers perceive that MDRO can affect nearly all aspects of the model. In contrast to "classical" patient-focused psychological and intervention models, systemic psychology focuses on the patient's environment and networks, especially the family. The Calgary model subdivides these aspects into three main themes: structural, functional, and developmental. Our findings suggest, that an MDRO diagnose can have a significant impact on the entire familiar system: An MDRO diagnose paired with insufficient information from medical professionals and the media ("*We had the Killer-Klebsiella*") suggests "danger" somewhat comparable to pandemic threats like Ebola, leading them to being cut off from their family systems.

This can occur because of the patient's own wishes ("*I do not want to endanger my family*"), or the wishes of the family members to protect the family or both. Furthermore, our codes revealed interference with family functioning, like daily activities ("*What do we do with the kids at home*?") and interaction with wider systems like neighbors ("*Everybody in the town knows about my situation*"). Isolation therefore seems to be a problem for both for the patient and their family. Considering that our discussion groups were not intended to evaluate whether general stigmatization, social tension [41] and prejudice is transferable from individuals to social groups (families) to organizations (hospitals, corporations) and to socio-political constructs like cities and even nations: epidemic outbreaks in the past have demonstrated that patients get separated from their families, families from their villages, villages from their countries and countries from the rest of the world: SARS, COVID-19 and Ebola [42] are merely recent examples.

## 3. Medical ethics

Taking the bio-ethical approach of four principles [30], especially the items "benevolence" and "non-maleficence", we observed a major interference with MDRO management. Patients with MDRO are usually (following national guidelines) segregated from others to keep other from being MDRO-colonized and infected [43]. Especially in persons (lay and staff) with limited knowledge (as participants mentioned several times here and in our preceding study) managing MDRO is equated with highly contagious diseases. Taking a utilitarian approach [44], an MDRO-infected patient's segregation is justified to protect the majority from the infected minority—putting MDRO-patients at risk for worsening social, economic and health-care related vulnerability. People perceive MDROs as a kind of pestilence. However, our findings only reveal bio-ethics and there is need for more investigation and work in this topic.

## Limitations

Our findings may be limited by the following factors:

Our evaluation concentrated on the subjective perceptions and experiences of health care providers working in intensive care settings. We did not assess the perceptions of patients and family members. Thus, it is unclear, whether our findings reflect the perceptions of patients and their families or staff from other specialties. On the other hand, our findings may be useful for exactly that subgroup, as they concern patients and relatives already burdened with a critical medical condition. Our findings may also be biased by individual exceptional experiences. As emotion cause humans to be highly attentive, leading them to recall memories better [11]. Health care providers' reports may just be "the tip of the iceberg" concerning their memories about MDROs. Therefore, the generalizability of their memories should be evaluated in other settings, e.g., in direct observation studies. Most of our codes concern socially significant isolation measures primarily and not the individual management of MDROs, like antibiotic treatments or surgery. Further studies should differentiate between MDROs themselves and MDRO management.

Over 50% of the coders in our first study also worked as coders in this follow up study. On the one hand, this confirms the grounded theory approach and on the other hand, fixation errors may have occurred. To avoid this, we decided to take a four-coder approach in contrast to the classical one or two-coder approaches. Additionally, we used a set of different frameworks (Atlas of Emotions, Beauchamps model, Calgary Model) for secondary deductive coding and not an inductively developed theme composition. We decided to use these well-established frameworks to enable better comparison and precise planning of ongoing research projects. We expect that where these frameworks interact (e.g., how emotions can affect ethical

decision making with regard to MDRO) may uncover especially important information concerning MDRO management and the medical education of health care workers. Selection bias may be present but is limited by this evaluation by a group of 11 professionals from different hospitals. Nevertheless, a broad approach, e.g., involving international online-surveys and different populations (medical staff, visitors, unaffected persons, affected persons and their families) should be taken to expand upon our results.

Furthermore, we did not examine whether patients and their relatives by experiencing MDRO loose trust hospitals or even whole health care systems in [45, 46] due to their emotional reactions. Thus, future research could concentrate whether MDRO management in hospitals causes patients to loose trust in medical systems and if this distrust causes people to avoid hospitals or health care providers altogether.

Last, future investigations could employ other methodologies, i.e., audio or videorecording to obtain deeper insight in participants' para- and nonverbal expressions, thereby revealing more information. At this stage of our investigation, we did not attempt that because of ethical considerations, but we will do so in future projects.

Additionally, we did not ask health care workers how long they spent talking to patients talking about MDRO or how they compensated for inadequate information or emotional crises. Nor did we determine whether care givers were trained in communication methods to help with strong emotions [47, 48]. We also did not assess whether time shortage and medical education were objectively or only subjectively perceived.

Next, the extent to which patients poorly informed about MDROs seek more information in the media and internet remains unclear to even more confusion, conflicts and impaired of the patient-physician-relationship as reported for dermatology [49].

All these limitations reveal this topics' broad spectrum and the great need for qualitative and quantitative future investigations in different settings (intensive care, rehabilitation, family medicine, emergency medicine and home care) and populations (health care workers, social workers, educators, lay persons, family members with and without MDRO experience).

Generalizing our findings (with this study affirming our first one's results), profound and ongoing multiprofessional medical education in MDRO management, perception and identification of emotional reactions known as "*empathic accuracy*" [17], communication techniques how to break bad news [50] and how to react to emotional reactions of oneself and of others [13] is needed to manage social systems in health care dealing with MDRO.

## Conclusion

Our article reports on the spread of MDRO on the emotional, ethical, and psycho-social perceptions of health care providers in different professions participating in an antimicrobial stewardship program. Our results confirm the findings we obtained in our first study: According to the perceptions of health care providers, MDROs evoke strongly emotional reactions in patients and staff. These may are attributable to insufficient medical education, communication errors and inconsistent information triggering emotional reactions, as well as over- and undertreatment.

Our results also generate new hypotheses for further research not only in workplace psychology. Furthermore, our findings raise ethical and family-psychological and sociologic questions that reflect the need for deeper research and reflection about how barrier precautions harm patients and family systems and how medical management (e.g., isolation) can be justified to protect others. Future research efforts, including the assessment of families and patients, should concentrate on our multi-facetted findings to generate a more holistic view of MDROs and barrier precautions and their ramifications on human life and social systems.

## Supporting information

**S1 Data. Data supplemental MDRO English version.**
(XLSX)

**S1 File.**
(DOCX)

## Acknowledgments

We thank all our study participants for their contribution. We also thank the Master of Medical Education Cohort 13 for their ongoing support in the didactic design of this Project, and are grateful for the logistical support provided by Andrea Rehberger, Sigrid Lemke and Veronica Koestermenke. We also thank Professor Michaela Riediger (Friedrich-Schiller University, Jena) for advice on emotional coding.

## Author Contributions

**Conceptualization:** Stefan Bushuven, Andreas Dietz.

**Investigation:** Stefan Bushuven, Stefanie Bushuven, Petra Dierenbach, Matthias Beiner.

**Methodology:** Stefan Bushuven, Thorsten Langer.

**Project administration:** Stefan Bushuven.

**Resources:** Stefan Bushuven.

**Supervision:** Markus Dettenkofer, Julia Inthorn, Thorsten Langer.

**Validation:** Stefan Bushuven, Thorsten Langer.

**Writing – original draft:** Stefan Bushuven, Markus Dettenkofer, Julia Inthorn, Thorsten Langer.

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
