## [Decision Letter · Decision Letter 0]

7 Nov 2020

PONE-D-20-25927

Emotional, social and ethical effects of Multidrug-Resistant Organisms: A Qualitative Study

Psychological Effects of multidrug resistant organisms

PLOS ONE

Dear Dr. Bushuven,

Thank you for submitting your manuscript to PLOS ONE. After careful consideration, we feel that it has merit but does not fully meet PLOS ONE’s publication criteria as it currently stands. Therefore, we invite you to submit a revised version of the manuscript that addresses the points raised during the review process.

We look forward to receiving your revised manuscript.

Kind regards,

Vijayaprasad Gopichandran

Academic Editor

PLOS ONE

Journal Requirements:

2.Thank you for including your ethics statement:  "Approval of the ethics board of the physician association Stuttgart was obtained prior to the investigation".   

We noted that in the online submission form, you indicated that ethical approval was not needed.

We understand that the framework for ethical oversight requirements for studies of this type may differ depending on the setting and we would appreciate some further clarification regarding your research. Could you please provide further details on whether approval was waived or exempted,, why your study is exempt from the need for approval and confirmation from your institutional review board or research ethics committee (e.g., in the form of a letter or email correspondence) that ethics review was not necessary for this study? Please include a copy of the correspondence as an "Other" file."

3. In your Methods section, please provide additional information about the participant recruitment method and the demographic details of your participants. Please ensure you have provided sufficient details to replicate the analyses such as: a) a description of any inclusion/exclusion criteria that were applied to participant recruitment, b) a table of relevant demographic details, c) a statement as to whether your sample can be considered representative of a larger population.

4. Please provide additional details regarding participant consent. In the ethics statement in the Methods and online submission information, please ensure that you have specified (1) whether consent was informed and (2) what type you obtained (for instance, written or verbal, and if verbal, how it was documented and witnessed). If your study included minors, state whether you obtained consent from parents or guardians. If the need for consent was waived by the ethics committee, please include this information.

5. We noted that you refer to this study as a cross sectional study in the manuscript, but according to your description we would not consider this a cross-sectional study but rather a qualitative study. In order to avoid confusion we would suggest that you change the wording in your manuscript and avoid referring to this study as a cross-sectional study.

6.Thank you for stating the following in the Acknowledgments Section of your manuscript:

[We thank all participants for their contribution and the Messmer Foundation, Radolfzell, Gemany for

financial support.]

 [The funders had no role in study design, data collection and analysis, decision to publish, or preparation of the manuscript.]

7. Please upload a copy of Figure 5, to which you refer in your text on page 17. If the figure is no longer to be included as part of the submission please remove all reference to it within the text.

Reviewers' comments:

Reviewer's Responses to Questions

**Comments to the Author**

1. Is the manuscript technically sound, and do the data support the conclusions?

Reviewer #1: Partly

Reviewer #2: Partly

Reviewer #3: Partly

2. Has the statistical analysis been performed appropriately and rigorously? 

Reviewer #1: I Don't Know

Reviewer #2: I Don't Know

Reviewer #3: Yes

3. Have the authors made all data underlying the findings in their manuscript fully available?

Reviewer #1: Yes

Reviewer #2: Yes

Reviewer #3: Yes

4. Is the manuscript presented in an intelligible fashion and written in standard English?

Reviewer #1: Yes

Reviewer #2: Yes

Reviewer #3: No

5. Review Comments to the Author

Reviewer #1: To the authors,

I'm particularly susceptible to this topic which is of interest in an era where ID is now a major threat due to covid, and might become the same if we dont fight antibiotic resistance. Better knowning the fear engendered by MDRO is very important to initiate acceptable measures and precautions.

Some minor corrections:

Introduction there is a gerrman typo "und separation" instead of and

"described by Paul Ekman" replace by Ekman et al.

Do not number from 1 to 6 if really relevant use i) etc in the text without spaces.

Main theme 2 and 3, remove BOLD characters

Major suggestions:

In the introduction and the discussion you are missing a recent article on the same topic by Hereng et al. that is close to your work and main message "Evaluation in general practice of the patient's feelings about a recent hospitalization and isolation for a multidrug-resistant infection" https://pubmed.ncbi.nlm.nih.gov/31047690/

It should be added.

For the coding process, can you add the percentage of the different feelings ? it will be easier to read

I don't feel that "experience of medical incompetence" is correct. It is not truly medical incompetence but maybe more the lack of formation and skills in the management of MDRO. Rephrase or clarify.

In the discussion of chapter 2 remove this paragraph that brings a conclusion meanwhile discussion should not conclude. "Conclusively, basing on the perception of the participants, not only microbes spread in human societies: emotions also do, contributing to a paralysis of family functioning and exclusion of patient from their social background."

Limitations : merge all the paragraph and remove the bold symbols. Also don't say first second etc until seventh, please classify logically and embed in the text the relevant data.

Conclusion could be improved. Don't say to our knowledge in a conclusion it's place is in the discussion section. Conclusion is factual. Same goes for study in Germany. It would not change it data where from spain for instance. You must make a generalizable summary.

For instance "Our work reports impact of MDRO carriage on emotional, ethical and family-psychological effects experienced by multiprofessional healthcare workers.

Then add your major findings. You cannot say that your finding are concordant wth your data ! "

Our results could confirm findings in our first study" weird way of making conclusion of new data.

Add what is clear in the abstract section instead (Conclusion: MDRO are perceived to have severe impact on emotions and affect bioethical and family psychological issues. Thus, further work should concentrate on these findings to generate a holistic view of MDRO on human life and social systems.)

Reviewer #2: In this paper the authors focus on the effects that MDRO patient-isolation protocols have on on healthcare workers; in particular, in the healthcare workers’ emotional experience after interacting with these patients and their families and some ethical implications. This is a very important area of research. However, I have some concerns regarding the methodology and conclusions provided in this study, please see below.

General Aim

It is not clear that the authors are actually reporting on the emotional, psychological and ethical effects on patients, as stated in the Introduction, Abstract and suggested in the title. This study is aimed at the experience of healthcare workers regarding the management of patients who are affected by MDRO and their families. As such, this must be clear from the beginning (rather than just being acknowledge as a limitation at the end of the paper).

Conclusion

The conclusion regarding “MDRO impairs Family Functioning” is interesting but requires more work with the patient and families themselves (rather than healthcare workers) to be able to claim the conclusions presented.

The “ 3. Medical Ethics” section needs work. The two principles used, benevolence and non-maleficence, are important and complex principles and it is not clear how they play a role in this issue. If this is an area of study they would like to include in their project (as indicated in the Introduction), they must attempt to understand the role that these principles play in the isolation protocols analyzed and the experiences of the people affected by those protocols, for example, questions such as: is the creation of these isolation protocols justified/or not by the benevolence principle? More analysis and discussion are needed to understand the role that healthcare workers’ perceptions play in an ethical analysis of this situation. Also, if the authors are not going to use Kant’s first formulation of the categorical imperative or a virtue ethics approach, it is not clear why are they mentioning it in the paper.

I would like to see the authors discuss in more detail their results in the context of the previous literature, how to they compare to previous findings (besides their own previous study)

Methods

I understand that for both deductive approaches, two coders had to agree to assign a code; however, I would like to see more detail, in particular regarding the reliability among coders. This is applicable to the other kinds of coding used in the paper, I would like to see more detail about the coder such as the training that the coders received and reliability achieved.

As noted in the limitations, 50% of the same coders as the previous study were used, that suggests that the coders are familiar with the hypothesis and somewhat biased. I am not satisfied with the explanation provided regarding the advantages of using theses coders. Perhaps if included more information on the coders (as stated above) it may somewhat mitigate this issue (although not resolved it entirely).

Finally, I agree with all the seven plus limitations mentioned for the authors, I had very similar concerns while reading this study. I am not entirely satisfied with the explanations they provide and would like to suggest that they add some methodological measurements to address these limitations. For example, as they acknowledge, their study focuses on “our evaluation concentrated on the subjective perceptions and experiences of health care providers from intensive care settings”, I would advise that they conduct interviews with the patients and families and correlate with eh data they already have.

Small Notes

Please proofread the paper for typos, such as the one on page 2 (Introduction) “und” should be replaced by “and”.

Additional data is presented in German if intended for an English publication, the authors may consider translating these additional data.

Reviewer #3: The study is well-made, but what concerns me is the syntax and incorrect structure of the language. In addition, there is vagueness in regards to the previous study which is not mentioned separately. In addition, the paper is subjective in nature and has a scope of improvement. Objectivity is the key for your study.

6. PLOS authors have the option to publish the peer review history of their article (what does this mean?). If published, this will include your full peer review and any attached files.

Reviewer #1: **Yes: **Benjamin Davido

Reviewer #2: No

Reviewer #3: **Yes: **NAVELI SHARMA

---

## [Author Response · Author response to Decision Letter 0]

25 Jan 2021

Stefan Bushuven MD MME

1. Style requirements

We checked the PLOS ONE templates and made appropriate changes:

- Vancover formatting style assigned instead of Numbered 

- Page numbers and Line numbers 

- Complete re-editing by a native speaking philologic professional with focus on idiomatic translation of the items 

2. Ethics statement

We included the ethical statement by the ethical board of the LANDESAERZTEKAMMER BADEN-WURTTEMBERG. Due to our anonymous data obtained, further ethical approval was considered unneeded. To ensure autonomy of the participates we consulted the workers’s council of the hospitals for additional validation and approval. 

We included the two documents in the re-submission. 

3. Demographic details of the participants 

We made major changes in the methods sections. We displayed participant demographic details in the results sections. 

Regarding the preceding studies with similar approach, we did not obtain further demographic data of our participants (like age or gender). Representability was assumed regarding the fact, that motivation to participate was not participation in a focus group as a science project, but an education lesson. This shows realistic populations of the interprofessional post - graduate formats and professionals working at our patients. However, selection bias and reduced representability cannot be excluded completely. This should be addressed with quantitative methods and analysis in special groups focusing on distinct age or profession.

4. Informed consent 

We made some correction in the methods sections and included the information letter for participation. Information was provided in written form. Consent was obtained verbally at the beginning and at the end of the lessons. Written consent was not obtained to guarantee anonymity. Minors were not part of our study. They would not have been excluded, but all post-graduate medical specialists were adults. 

5. Study design 

We changed the study type from “cross sectional” to “qualitative”

6. Acknowledgments

We made a correction as desired. The Messmer Foundation only covers the publication costs only. The whole study itself was conducted without external financial support. 

7. Figure 5

This was a typo. We changed Figure 5 to 4.

8. Captions

We corrected the document concerning captions for supporting information

Corrections due to Reviewer #1

1. Minor corrections were considered and implemented.

2. The Article by Hereng was implemented into the introduction and discussion section.

3. We clarified the term of medical incompetence weakening it to the experience of fails. 

4. We removed the conclusive paragraph from Discussion Chapter 2 as suggested. 

Corrections due to Reviewer #2

1. Due to the suggestions, we changed the title addressing the perceptions and not per se the effects of MDRO for precision of the article headline. Additionally, we made some amendments to the introduction focusing on the same issue. 

2. We added the coders’ instruction process for clarification of the process. 

3. We added the number of codes 3 or 4 of the coders agreed on in table one, to address reliability of the code assignment. Indeed, some codes were “very strong”, as they were assigned by all the coders. However, all codes presented were accepted once by at least three of the coders. 

4. The bias provoked by two of the coders familiar with the topic was part of the study protocol to enhance validity. To our knowledge most qualitative studies are conducted be single or only two-coder approaches (Lavrakas 2008 and Riffe et al 2005). Thus, we considered our approach to be “above average”. 

5. We appreciate the proposal to do more qualitative work on family members and patients. But this was not the objective of the study. We are preparing these approaches to enhance validity and reliability in future projects. 

6. We deleted classical philosophically approaches within the paper like the categoric imperative. Indeed, it does not contribute to this kind the study or the study objectives. Further projects succeeding this study may focus on the bioethical, Kantian or utilitarian approaches. 

Corrections due to Reviewer #3

1. We reread the paper and made corrections according to the suggestions (typography, interpunctation) with assistance of the mentioned native speaker. See preceding comments by reviewer #1 und #2

2. The reviewer addresses the subjectivity rather than objectivity of the paper without going into detail. With further information in the analysis section, we hope to satisfyingly address these concerns. 

3. We presented the former study in the introduction section in deeper detail and intensively revised the introduction section especially regarding typography and style.

4. We changed “unempathy” to “lack of empathy” as suggested

5. We added the labelling of the axes in the figures

---

## [Editor Report · Decision Letter 1]

27 Jan 2021

Interprofessional Perceptions of  emotional, social and ethical effects of Multidrug-Resistant Organisms: A Qualitative Study

PONE-D-20-25927R1

Dear Dr. Bushuven,

We’re pleased to inform you that your manuscript has been judged scientifically suitable for publication and will be formally accepted for publication once it meets all outstanding technical requirements.

Kind regards,

Vijayaprasad Gopichandran

Academic Editor

PLOS ONE
---

## [Editor Report · Acceptance letter]

11 Feb 2021

PONE-D-20-25927R1 

Interprofessional Perceptions of emotional, social, and ethical effects of Multidrug-Resistant Organisms: A Qualitative Study 

Dear Dr. Bushuven:

I'm pleased to inform you that your manuscript has been deemed suitable for publication in PLOS ONE. Congratulations! Your manuscript is now with our production department. 

Kind regards, 

on behalf of

Dr. Vijayaprasad Gopichandran 

Academic Editor

PLOS ONE